# Satisfaction with COVID-19 Vaccines in Health Care Workers and the General Population: A Cross-Sectional Study in Urban Bangkok, Thailand

**DOI:** 10.3390/vaccines10081345

**Published:** 2022-08-18

**Authors:** Jadsada Kunno, Pataraporn Yubonpunt, Chavanant Sumanasrethakul, Chuthamat Kaewchandee, Mark Gregory Robson, Wachiraporn Wanichnopparat, Krit Prasittichok, Titaporn Luangwilai, Chonlawat Chaichan, Patcharaporn Krainara, Busaba Supawattanabodee

**Affiliations:** 1Department of Research and Medical Innovation, Faculty of Medicine Vajira Hospital, Navamindradhiraj University, Bangkok 10300, Thailand; 2Department of Public Health, Faculty of Public and Environmental Health, Huachiew Chalermprakiet University, Samut Prakan 10540, Thailand; 3Department of Urban Medicine, Faculty of Medicine Vajira Hospital, Navamindradhiraj University, Bangkok 10300, Thailand; 4School of Environmental and Biological Sciences, Rutgers University, New Brunswick, NJ 08901, USA

**Keywords:** COVID-19 vaccine, vaccine satisfaction, general population, healthcare workers, Bangkok

## Abstract

Background: COVID-19 vaccination hesitancy is a global issue. Many people are concerned about experiencing side effects from the vaccine. This study evaluated satisfaction with the COVID-19 vaccine in the general population (GP) and healthcare workers (HCWs) in Bangkok, Thailand. Methods: A cross-sectional online survey was distributed from September-December 2021. Independent sample *t*-tests were used to compare GP and HCW participants’ total vaccine satisfaction scores as well as their satisfaction with varying vaccine types. Multiple linear regression was used to identify predictors of satisfaction scores among GP and HCWs. Results: A total of 780 valid questionnaire responses were obtained. The majority of GP participants (n = 390) had received their first (93.3%) and second (88.5%) vaccination shots by viral vector vaccine; however, 90% had not received a third dose (booster). In contrast, the majority of HCW participants (n = 390) had received their first (92.8%) and second (82.8%) vaccination doses by the inactivated vaccine, and 83% had received a third vaccine dose. HCWs had significantly higher total satisfaction scores than GP participants (*p* = 0.034), and they were also significantly more satisfied with the mRNA vaccine as a third dose (*p* = 0.001). Multiple linear regression models found less association with vaccine satisfaction among GP participants who had not isolated following exposure to COVID-19 and those who have never been at risk of infection (ᵦ −0.159; 95% CI −12.867, −1.877; *p* = 0.009). Among HCWs, being married (ᵦ 0.157; 95% CI 0.794, 3.278; *p* = 0.001) or divorced (ᵦ 0.198; 95% CI 3.303, 9.596; *p* < 0.01) was more closely associated with vaccine satisfaction than being single. Conclusion: HCWs were more satisfied with the type and efficacy of inactivated, viral vector, and mRNA vaccines than GP participants, and the former were also more satisfied with the cost of vaccine boosters. Our results indicate that satisfaction with the COVID-19 vaccine is based on academic knowledge sharing and the government’s promotion efforts. Future research will explore strategies to raise awareness about the importance of vaccination.

## 1. Introduction

Coronavirus disease 2019 (COVID-19) is an infectious disease caused by a newly discovered coronavirus. In Thailand, COVID-19 outbreaks have been traced to superspreading events at sites that attracted crowding, extended interactions, and high turnover, particularly entertainment establishments such as pubs, bars, Karaoke lounges, and various types of gambling venues [1]. Such events have led to the spread of COVID-19 to many provinces. Currently, the Thai government is promoting vaccination and distributing vaccines to the whole population.

Vaccination is critical for infection control and prevention. Currently available vaccines against severe acute respiratory syndrome coronavirus-2 (SARS-CoV-2) include messenger ribonucleic acid (mRNA)-based, viral vector-based, protein subunit, and whole virus or “inactivated virus” vaccines [2,3], However, many people are concerned about the potential side effects of these vaccines; vaccine hesitancy and refusal have emerged as substantial barriers to combatting the pandemic. For example, studies have found that some individuals oppose vaccines due to a preference for “natural” rather than “artificial” drugs, whereas others cite strong religious beliefs [4,5].

COVID-19 vaccine boosters are third doses of two-dose vaccines such as mRNA vaccines or second doses of single-dose vaccines such as viral vector-based vaccines [6]. In Thailand, 75% of the population has received a single dose vaccine, 69% a second dose booster, and 18.8% have received a third dose booster [7]. However, pandemic outbreaks and vaccine inaccessibility are major challenges to achieving 100% vaccination rates. In addition, distrust of vaccine boosters has been triggered by anti-vaccination campaigns and is becoming increasingly prevalent [8]. Epidemiologists and public health researchers are experiencing challenges in their efforts to provide evidence-informed recommendations regarding COVID-19 vaccine booster timing and promote full vaccination to priority groups [6]. Individuals who have already been vaccinated may accept the side effects they experienced following the perceived ineffectiveness of the vaccine booster dose, susceptibility to the target infection, and safety uncertainties [9,10].

Factors influencing vaccine satisfaction included the sociodemographic characteristics of the individuals, and social and organizational factors [11]. In addition, certain characteristics of COVID-19 vaccines themselves influenced public attitudes towards accepting the vaccines. Attitudes towards vaccines are influenced by a complex interaction of various social, cultural, political, and personal factors, including doubts and hesitation about vaccination [12]. It is widely believed that factors such as risk perception beliefs have a substantial effect on the intention to be vaccinated [13].

To date, few studies have focused on satisfaction with the COVID-19 vaccine booster [14,15,16]. Previous studies have reported that the most common reasons cited for refusing the vaccine were insufficient knowledge about potential negative long-term effects, distrust of drug companies’ assurances that the vaccine is safe, a belief that the virus is not dangerous, and doubts concerning the short-term dangers of vaccines [14]. In addition, some researchers have questioned if healthcare workers should receive priority access to vaccines [15]. Front-line healthcare workers have at least a threefold increased risk of being infected with COVID-19 compared with the general population, even after accounting for other risk factors [16].

The general population (GP) and healthcare workers (HCWs) perception of the vaccine booster might be affected by many things such as positive or negative feelings that refer to satisfaction and are based on the sociodemographic characteristics of the individuals. Thus, this study evaluated satisfaction with COVID-19 vaccines and boosters among members of the GP and HCWs. We examined the effect of socio-demographic factors such as age, gender, status, education, occupation, risk of contracting COVID-19, vaccine, and booster history in each group. To our knowledge, this is the only study to examine both of these groups in the urban community of Bangkok, Thailand.

## 2. Methods

### 2.1. Study Design

This study entailed the analysis of a cross-sectional survey distributed among GP participants and HCWs in urban Bangkok, Thailand from September-December 2021. The study was approved by the ethics committee of the Faculty of Medicine, Vajira Hospital, Navamindradhiraj University, Bangkok, Thailand. The Institutional Review Board of the Faculty of Medicine Vajira Hospital is in full compliance with the international guidelines for human research protection as stated in the Declaration of Helsinki, the Belmont Report, Council for International Organizations of Medical Sciences (CIOMS) Guidelines, and the International Conference on Harmonization in Good Clinical Practice (ICH-GCP). (COA 151/2564). All experiments were performed in accordance with relevant guidelines and regulations, in the Declarations section.

### 2.2. Participants

Participants aged 18 years and older living in Bangkok (n = 780), were eligible to participate in the study. The invitation asked participants to confirm that informed consent was obtained from all subjects, voluntary participation, and provided instructions for filling in the questionnaire. The sample size was calculated using G*Power based on the estimated population of GP participants and HCWs in the city (Figure 1).

### 2.3. Data Collection

Data were collected using an online survey (Google form, including consent form) that was distributed on social media using a snowball technique. The invitation asked participants to confirm that informed consent was obtained from all subjects, voluntary participation, and provided instructions for filling in the questionnaire.

### 2.4. Questionnaire

The questionnaire takes about 10 min to complete and is divided into two sections. The questions were designed and modified by an expert team of researchers. The first section collected socio-demographic information, including age (years), sex, status, education, occupation, risk of contracting COVID-19, and vaccination booster history (see Table 1 for details). The second section consisted of 15 items measuring satisfaction, which were scored on a 3-point Likert scale ranging from High satisfaction = 3, Medium satisfaction = 2, and Low satisfaction = 1 (see Table 2 for details). The satisfaction item responses were summed to a total score ranging from 0–45 and devised into three levels: group 1 scores ranging from 15–25 denoting low satisfaction; group 2 scores ranging from 26–35 indicating medium satisfaction; and group 3 scores ranging from 36–45 representing high satisfaction. Content validity was examined by three experts, and the item objective congruence (IOC) score was 0.93. This study reported a Cronbach’s alpha coefficient of 0.751.

### 2.5. Statistical Analysis

Descriptive statistics, including frequency, percentages, mean, and standard deviations, were analyzed to describe GP participants and HCW’s socio-demographic characteristics and satisfaction item scores. Independent sample *t*-tests were used to compare the influence of vaccination booster types on satisfaction scores and compare satisfaction scores between GP and HCWs. Multiple linear regression was used to identify predictors of satisfaction scores among GP and HCWs. Statistical analysis was performed using the Statistical Package for the Social Sciences Program (Statistical Package for the Social Sciences Program (SPSS) version 22 (IBM Corp., Armonk, NY, USA). The level of statistical significance was considered at *p* < 0.05.

## 3. Results

### 3.1. Participant Characteristics

A total of 780 questionnaire responses were obtained, including 390 general population (GP) participants and 390 healthcare worker (HCWs) participants. Socio-demographic details are presented in Table 1.

The mean age of the GP participants was 55.14 ± 17.35 years. In this group, 54.9% were female, 51.0% were married, 43.3% had a bachelor’s degree, 35.2% worked in government offices, and 70% had never been at risk of contracting COVID-19. The majority of GP participants had received their first (93.3%) and second (88.5%) vaccine doses by viral vector vaccine. However, 89.7% reported that they had not received a third or booster dose.

The mean age of the HCWs was 35.13 ± 11.28 years. In this group, 96.9% were female, 68.5% were single, and 46.4% reported having isolated due to exposure to COVID-19 infection. The majority of HCW participants had received their first (92.8%) and second (82.8%) vaccine doses by inactivated vaccine. Most HCWs (83.6%) reported that they had received a third or booster shot.

### 3.2. Vaccine Satisfaction Scores

Participants’ vaccine satisfaction scores are summarized in Table 2. The descriptive statistics analysis shows low-to-medium satisfaction for all vaccine types. None of the scores rose much higher than 2.0, which denotes medium satisfaction: “from High satisfaction = 3, Medium satisfaction = 2, and Low satisfaction = 1”.

Both GP participants and HCWs were more satisfied with the viral vector vaccine (2.14 ± 1.00 and 1.83 ± 0.63, respectively) and the inactivated vaccine. HCWs scored higher than GP participants on the item measuring knowledge and understanding of vaccines that have been determined to be of good quality (2.05 ± 0.65 and 1.91 ± 0.99); however, GP participants had greater confidence in the quality of available vaccines than HCWs (1.92 ± 1.05 and 1.65 ± 0.65).

In terms of vaccination booster management, GP participants had greater satisfaction than HCWs that there was enough viral vector vaccine for the Thai population, that registration for injections of the viral vector vaccine would be conducted on time, vaccines were being distributed across all areas, and the government had credibility in procuring and administering vaccines. However, HCWs expressed greater satisfaction than GP participants that there was a sufficient inactivated vaccine for the Thai population and that registration for injections of the inactivated would be conducted on time.

In terms of satisfaction with vaccine boosters, HCWs were more satisfied than GP participants with the type and efficacy of the inactivated vaccine, viral vector vaccine, and mRNA vaccine as well as the vaccine booster cost.

### 3.3. Comparison of Total Vaccine Satisfaction Scores

Table 3 presents the result of the independent *t*-test comparing total vaccine satisfaction scores between GP and HCWs participants. The table shows that HCWs expressed significantly higher satisfaction with the COVID-19 vaccine booster than GP participants (*p* = 0.037).

### 3.4. Comparison of Satisfaction with Vaccine Types

Table 4 presents the results of independent *t*-tests comparing satisfaction with vaccine types among GP and HCWs participants. In the first dose category, we found no significant difference between the two groups’ satisfaction with the inactivated vaccine, viral vector vaccine, or mRNA vaccine.

In the second dose category, our results indicate no significant difference in GP and HCWs participants’ satisfaction with any of the vaccine types; however, the difference in satisfaction with the viral vector vaccine (*p* = 0.056) was nearly significant.

In the third dose category, the results indicate that HCWs expressed significantly greater satisfaction with the mRNA vaccine (*p* = 0.001). However, there was no significant difference in satisfaction with the other vaccine types.

### 3.5. Association between Participant Characteristics and Satisfaction with COVID-19 Vaccines

Table 5 shows the results of multiple linear regression modeling of the association between independent variables and vaccine satisfaction among GP and HCWs participants. Among GP participants, who had been exposed to the risk of contracting COVID-19 but had not self-isolated, was less associated with vaccine satisfaction than “never at risk” (ᵦ −0.159; 95% CI −12.867, −1.877; *p* = 0.009)

Among HCWs, being married (ᵦ 0.157; 95% CI 0.794, 3.278; *p* = 0.001) and or divorced (ᵦ 0.198; 95% CI 3.303, 9.596; *p* < 0.01) was more closely associated with vaccine satisfaction than being single. In addition, the risk of infection of contracting COVID-19 by not isolating (ᵦ −0.206; 95% CI −5.013, −0.242; *p* = 0.031) and those who had isolated after exposure to the COVID−19 (ᵦ −0.282; 95% CI −5.627, −0.990; *p* = 0.005) were less associated with vaccine satisfaction than “never at risk”. In addition, HCWs who received the 3rd dose of vaccination (ᵦ −0.844; 95% −1.177, −0.512; *p* < 0.001), were associated with vaccine satisfaction more than those that only received the 1st dose of vaccination.

## 4. Discussion

This study was conducted during the third wave of the COVID-19 pandemic amid the distribution of vaccine boosters in Thailand. To the best of our knowledge, it is the first study to evaluate satisfaction with the COVID-19 vaccine booster among both GP participants and HCWs in an urban community. We found that socio-demographic factors such as age, sex, status, education, occupation, risk of contracting COVID-19, and vaccine dose history influenced vaccine satisfaction. In addition, we found that HCWs were significantly more satisfied with vaccines overall as well as with mRNA vaccines as a third dose.

### 4.1. Vaccine Satisfaction Scores among GP and HCWs

We found that HCWs were more satisfied with the vector vaccine than GP participants and had good knowledge and understanding of vaccines that have been determined to be of good quality. This result can be compared with a previous study that reported high satisfaction with the safety and immunogenicity of inactivated COVID-19 vaccines among HCWs [17]. In that study, the overall incidence of adverse reactions to inactivated vaccines was 59.5%, and local adverse events occurred more frequently than systemic adverse events after the first dose (52.4% vs. 37.3%) [17]. HCPs with higher scores were associated with mandatory vaccination support (*p* < 0.001) [18,19]. The immune responses in people infected with the virus after vaccination, antibody development, and the course of the disease are still unclear [20]. In fact, the vector scores were higher using findings on the high percentage of adverse reactions as a comparison.

In addition, our results indicate that HCWs were more likely than GP participants to select a viral vector vaccine in the future if possible. We propose that HCWs were more satisfied with the viral vector vaccine because the immunogen is expressed in the context of a heterologous viral infection, which induces the innate immune responses required for the adaptive immune responses [3,21]. However, one study suggested that individuals should obtain a second dose of the inactivated vaccine along with possible booster doses [17]. Currently, most Thais have only received a single dose of inactivated vaccine. However, both GP and HCWs participants were more satisfied with the viral vector vaccine than the inactivated vaccine, and GP participants had higher confidence in the quality of the currently available vaccines than HCWs. We attribute this result to academic knowledge that viral vector vaccines might be more efficient than inactivated vaccines.

Our study found the majority of GP participants were satisfied that there were sufficient viral vector vaccine doses for the Thai population, registration for injections of viral vector vaccine doses would be conducted on time, vaccine allocation was distributed across all areas, and the government was credibly procuring and administering vaccines. These results suggest that the Thai government’s support for and promotion of vaccination is effective. The finding that HCWs were satisfied that there were sufficient inactivated vaccine doses for the Thai population and that registration for injections of inactivated vaccine doses would be conducted on time may be attributable to the fact that in Thailand, the first dose is the most common for the inactivated vaccine, followed by the vector vaccine.

HCWs were more satisfied with the type and efficacy of the inactivated, viral vector, and mRNA vaccines as well as the cost of vaccine boosters than GP participants. Our results align with those of a Czech study that reported a high level of vaccine booster acceptance (71.3%) among Czech HCWs and suggested that perceived safety and ethical dilemmas of vaccine justice should be addressed while communicating with HCWs and other population groups [6].

Our study suggested that satisfaction with COVID-19 vaccines among GP and HCWs participants was based on academic knowledge sharing and government support to promote vaccination for the entire Thai population. A previous study found that 50% vaccine efficacy was associated with a 51% rate of acceptance [22]. As our results show, vaccine willingness or hesitancy may vary depending on some, but not all, of these attributes. It is important that healthcare professionals and representative groups, should be closely involved in policymaker and health authority decisions regarding the establishment and implementation of vaccine recommendations and interventions to address vaccine hesitancy [23].

### 4.2. Comparison of Total Vaccine Satisfaction Scores

Our finding that HCWs had significantly higher total vaccine satisfaction scores than GP participants is similar to the results of other studies reporting that HCWs can improve their perceptions and acceptance of vaccination [24,25]. One study found that only 20% and 24% of HCWs respondents preferred to receive the viral vector vaccine or the mRNA vaccine, respectively [24]. Another study reported that HCWs demonstrated significantly more willing to undergo vaccination than the general population [26]. Public vaccinations in Japan started with medical workers (about 4.7 million people) in February 2021, followed by people 65 years or older (about 36 million people) in April 2021 [27]. Although around 30% of the Japanese are unsure or neutral regarding vaccine acceptance, the government’s vaccine campaign sought to promote vaccination for all people [27]. However, higher vaccine satisfaction among HCWs may be due to their greater risk of infection with COVID-19.

### 4.3. Comparison of Satisfaction with Vaccine Types

Our study is the first to compare satisfaction with varying vaccine types among GP and HCWs in Bangkok, Thailand. We found no significant difference between the two groups’ satisfaction with any of the vaccine types for the first dose. Our finding that all participants had received at least one vaccine dose attests to the effectiveness of the Thai government’s support in promoting vaccination, and it is similar to a previous study that found that only 0.6% of participants had not received their first COVID-19 vaccine dose [28].

Our finding of a nearly significant difference (*p* = 0.056) between GP and HCWs participants’ satisfaction with viral vector vaccines as a booster aligns with a previous study reporting that 55.3% of participants were willing to receive a vaccine booster [28]. Although there was no significant difference in the two groups’ satisfaction with inactivated vaccines and mRNA vaccines as a second dose, participants in another study expressed the perceptions that the mRNA vaccine is effective against COVID-19 variants (*p* < 0.001) and that mixing/matching vaccines is effective against variants (*p* < 0.001) [28].

We found that HCWs expressed significantly greater satisfaction with the mRNA vaccine as a third/booster dose than GP participants (*p* < 0.01). A previous study indicated that perceptions of the effectiveness of the viral vector and mRNA vaccines and mixing both vaccines against the Delta variant all correlated significantly and positively with each other (*p* < 0.01) [28]. However, many more HCWs than GP participants had received a third vaccine dose. One study found that 77.8% of respondents agreed that a booster dose would be needed [20]. However, for people with a high level of stress, other health programs need to be developed to enhance their positive attitude toward the COVID-19 vaccine [29].

A recent study showed that 75% of the Thai population have received a first vaccine dose, 69.5% have obtained a second dose, and 18.8% have had a third dose [7]. Another study showed that 70% intended to be vaccinated when the COVID-19 vaccine was approved under Emergency Use Administration [30]. Hence, we recommend that public health officials scale up efforts to disseminate reliable information about the different COVID-19 variants and heavily promote the vaccine booster. Further research on methods to alleviate worries about emerging variants is warranted.

### 4.4. Association between Participant Characteristics and Vaccine Satisfaction

Our results showed that among GP and HCWs participants, who had been exposed to the risk of contracting COVID-19 but had not self-isolated, was less associated with vaccine satisfaction than “never at risk,” which might be related to personal awareness. Additionally, social determinants of health such as gender, age, number of children in the family, and the degree of satisfaction with life were also predictors (*p* < 0.05) for COVID-19 vaccine acceptance [31]. Protecting oneself as well as others have been shown to be the main reason for willingness to be vaccinated, and concerns about serious side effects and the safety of vaccines have been shown to be the main reasons for unwillingness [32]. One study presented that satisfaction with vaccine-related promotional materials was determinant of behavioral intention [33]. However, our results showed among HCWs participants, those who had isolated after exposure to COVID-19 were also less associated with vaccine satisfaction. They rated fear of side effects (87.18%) and lack of information (70.94%) as the reasons for the hesitancy [34].

Our finding of a closer association between married and divorced status and vaccine satisfaction and being single among HCWs agrees with the results of previous studies reporting that married participants were more likely to accept the vaccine [27,35], and that divorce is a major life stressor that may cause at least short-term mental distress, in part due to the emotional and financial burdens, which might contribute to higher vaccine acceptance [36]. Future studies could also incorporate participants’ perceptions of the current situation. One study suggested that green pass implementation must be accompanied by effective education and information strategies for the target population [37,38].

## 5. Limitations

This study was limited by a number of factors. For one, it was conducted online using self-reported data, which could introduce selection bias. Therefore, this sampling method can have a potential sampling bias, margin of error, based on the comparison between two markedly different populations. In addition, for this study, the *t*-test analysis used might lead to spurious conclusions. Finally, our study was unbalanced in terms of sex distribution which could introduce conclusion bias.

## 6. Conclusions

This study explored GP and HCWs participants’ satisfaction with COVID-19 vaccines during the third wave of the pandemic in Thailand. HCWs were more satisfied with the type and efficacy of inactivated, viral vector, and mRNA vaccines than GP participants, and they expressed greater satisfaction with the cost of vaccine boosters. Therefore, HCWs perceive the benefit of vaccinations despite the potential side effects; however, risk perceptions and discounting the future benefits could dissuade some members of the general population from getting vaccinated. However, satisfaction with COVID-19 vaccines and boosters among members of GP and HCWs might be based on individual factors such as sex, status, education, and risk of contracting COVID-19. Thus, this study is based on the current situation of GP and HCWs using COVID-19 vaccines and boosters in Bangkok Thailand. It is important to monitor changes in vaccine acceptance as the vaccine development process continues. Future research will explore effective strategies to raise awareness about the importance of vaccination.

## Figures and Tables

**Figure 1 vaccines-10-01345-f001:**
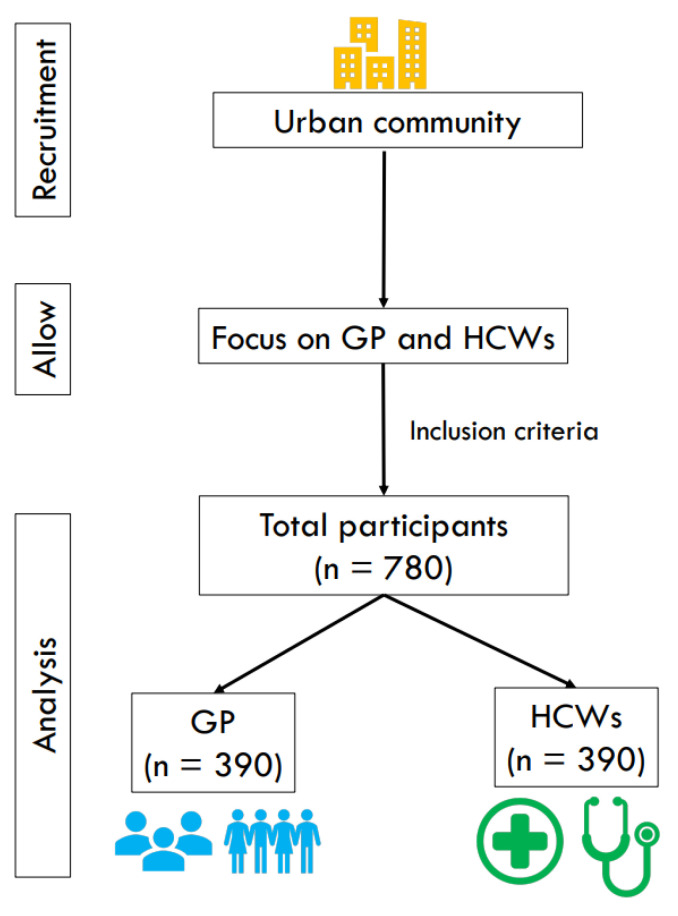
Flowchart of COVID-19 Vaccine Survey among GP and HCWs (n = 708).

**Table 1 vaccines-10-01345-t001:** Characteristics of participants (n = 780).

	Sociodemographic	GP N = 390	HCWs N = 390
		n (%) or median ± SD	n (%) or median ± SD
Age (Years)		55.14 ± 17.35	35.13 ± 11.28
Sex			
	Male	179(45.1)	12(3.1)
	Female	214(54.9)	378(96.9)
Status			
	Single	132(33.8)	267(68.5)
	Married	199(51.1)	110(28.2)
	Divorced	59(15.1)	13(3.3)
Education			
	Primary school	36(9.3)	1(0.3)
	High school	70(17.9)	18(4.6)
	Diploma	62(15.9)	28(7.2)
	Bachelors	169(43.3)	307(78.7)
	Above Bachelors	53(13.6)	36(9.2)
Occupation			
	HCWs	-	390(100)
	Government office	137(35.2)	-
	Private office	70(17.9)	-
	Non-office employee	32(8.2)	-
	Entrepreneur	75(19.2)	-
	Housewife	76(19.5)	-
Risk of contracting COVID-19			
	Risk exposure: isolation	34(8.7)	181(46.4)
	Risk exposure: no isolation	30(7.7)	117(30.0)
	Not sure	53(13.6)	26(6.7)
	Never at risk	273(70.0)	66(16.9)
Vaccination history			
1st dose			
	Inactivated vaccine	25(6.4)	324(83.1)
	Viral vector vaccine	364(93.3)	48(12.3)
	mRNA vaccine	1(0.3)	18(4.6)
2nd dose			
	None	19(4.9)	2(0.5)
	Inactivated vaccine	26(6.6)	323(82.8)
	Viral vector vaccine	345(88.5)	41(10.5)
	mRNA vaccine	0	24(6.2)
3rd dose			
	None	350(89.7)	64(16.4)
	Inactivated vaccine	15(3.8)	6(1.5)
	Viral vector vaccine	18(4.7)	102(26.2)
	mRNA vaccine	7(1.8)	218(55.9)

**Table 2 vaccines-10-01345-t002:** Descriptive statistics of satisfaction scores on COVID-19 vaccine among the general population and healthcare workers.

Satisfaction Items	GP	HCWs
	Median ± SD	Median ± SD
Vaccination booster type		
1. I am satisfied with inactivated vaccine.	1.25 ± 0.92	1.39 ± 0.60
2. I am satisfied with the viral vector vaccine.	2.14 ± 1.00	1.83 ± 0.63
3. I have confidence in the quality of the available vaccines.	1.92 ± 1.05	1.65 ± 0.65
4. I have a good knowledge and understanding of vaccines that have been determined to be of good quality.	1.91 ± 0.99	2.05 ± 0.65
5. If possible, I will select a viral vector vaccine	2.09 ± 1.14	2.14 ± 0.78
Vaccination booster management		
6. There are sufficient inactivated vaccine doses for the Thai population	1.52 ± 1.01	1.75 ± 0.73
7. There are sufficient viral vector vaccine doses for the Thai population	1.73 ± 1.01	1.60 ± 0.65
8. The registration for injection of inactivated vaccine will be conducted on time (No delay)	1.50 ± 1.00	1.72 ± 0.70
9. The registration for injection of viral vector vaccine will be conducted on time (No delay)	1.92 ± 1.08	1.71 ± 0.65
10. Each vaccine allocation is distributed across all areas.	1.52 ± 0.98	1.47 ± 0.60
11. The government’s credibility in procuring and administering vaccines	1.48 ± 1.16	1.43 ± 0.60
Satisfactions with vaccine booster		
12. I am satisfied with the type and efficacy of the inactivated vaccine.	1.58 ± 1.01	1.72 ± 0.63
13. I am satisfied with the type and efficacy of the viral vector vaccine	1.85 ± 1.11	2.50 ± 0.64
14. I am satisfied with the type and efficacy of the mRNA vaccine.	1.73 ± 1.09	2.37 ± 0.63
15. I am satisfied with the vaccine cost	1.45 ± 0.95	1.71 ± 0.63

Maximum score = 3.

**Table 3 vaccines-10-01345-t003:** Comparison of total vaccine satisfaction scores.

Population	Satisfactions on COVID-19 Vaccine Booster			
	Median ± SD	*t*	*p*	95% CI
GP	25.65 ± 12.36	−2.091	0.037	−2.80, −0.087
HCWs	27.10 ± 5.85			

Note: Maximum score = 45.

**Table 4 vaccines-10-01345-t004:** Comparison of satisfaction with vaccine types.

Vaccine Types		Satisfaction Scores				
		GP	HCWs	*t*	*p*	95% CI
		Median ± SD	Median ± SD			
1st dose vaccination						
	No vaccine	-	-	-	-	-
	Inactivated vaccine	26.16 ± 11.93	27.17 ± 5.79	−0.422	0.677	−5.97, 3.94
	Viral vector vaccine	25.65 ± 12.40	27.16 ± 5.80	−1.422	0.158	−3.60, 0.59
	mRNA vaccine	-	-	-	-	-
2nd dose vaccination booster						
	No vaccine	30.15 ± 9.63	27.50 ± 14.84	0.248	0.843	−109.63, 114.950
	Inactivated vaccine	26.11 ± 12.22	27.15 ± 5.75	−0.431	0.670	−6.015, 3.93
	Viral vector vaccine	25.37 ± 12.48	27.56 ± 5.83	−1.93	0.056	−4.43, 0.058
	mRNA vaccine	-	-	-	-	-
3rd dose vaccination booster						
	No vaccine	25.54 ± 12.42	27.03 ± 6.42	−1.422	0.157	−3.54, 0.57
	Inactivated vaccine	24.80 ± 14.87	29.00 ± 7.23	−0.867	0.398	−14.38, 5.98
	Viral vector vaccine	27.33 ± 11.76	30.28 ± 5.63	−1.043	0.310	−8.88, 2.98
	mRNA vaccine	28.42 ± 1.51	25.58 ± 5.12	4.256	0.001	1.37, 4.31

Maximum score = 45; Data were presented as t-test, and 95% confidence interval (CI).

**Table 5 vaccines-10-01345-t005:** Factors associated with satisfaction with COVID-19 vaccine booster among the general population and healthcare workers.

Variable		GP		HCWs	
		β (95% CI)	*p*	β (95% CI)	*p*
Sex					
	Male	Ref.		Ref.	
	Female	−0.025 (−3.167, 1.924)	0.631	−0.019 (−3.861, 2.605)	0.703
Status					
	Single	Ref.		Ref.	
	Married	0.047 (−1.653, 3.979)	0.417	0.157 (0.794, 3.278)	0.001
	Divorced	−0.068 (−6.252, 1.556)	0.238	0.198 (3.303, 9.596)	<0.001
Education					
	≥ Bachelor’s degree	Ref.		Ref.	
	< Bachelor’s degree	−0.042 (−3.59, 1.517)	0.424	0.070 (−0.479, 2.989)	0.155
Risk of contracting COVID-19					
	Never at risk	Ref.		Ref.	
	Not sure	−0.008 (−3.839, 3.406)	0.906	−0.031 (−3.028, 2.066)	0.711
	Risk exposure: no isolation	−0.159 (−12.867, −1.877)	0.009	−0.206 (−5.013, −0.242)	0.031
	Risk exposure: isolation	0.045 (−3.386, 7.334)	0.469	−0.282 (−5.627, −0.990)	0.005
Vaccination history					
	1st dose	Ref.		Ref.	
	2nd dose	−0.304 (−1.303, 0.694)	0.549	−0.510 (−1.391, 0.371)	0.256
	3rd dose	0.180 (−0.645, 1.005)	0.668	−0.844 (−1.177, −0.512)	<0.001

Data were presented as ᵦ coefficients and 95% confidence interval (CI).

## Data Availability

The data sets generated and analyzed during the current study are not publicly available due to identifiable information but are available from the corresponding author on reasonable request answering the survey.

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
