# Peer review of "Satisfaction with COVID-19 Vaccines in Health Care Workers and the General Population: A Cross-Sectional Study in Urban Bangkok, Thailand"

_vaccines, 2022, doi:10.3390/vaccines10081345_

Round 1
Reviewer 1 Report
The manuscript is intended to describe satisfaction with COVID-19 vaccines and boosters 129 among members of general population (GP) and healthcare workers (HCWs).In its current version, the study presents some flaws that must be addressed before it could be considered worthy of publication on Vaccines.
Following are revisions needed:
Introduction
The introduction should specify the theme, the context, the justification, and the hypotheses of the research. This introductory part should end with the main objective of the study.
The authors stated: “To-date, few studies have focused on satisfaction with the COVID-19 vaccine booster.”. Please, cite these studies.
Methods
It is not given any information regarding a pilot study for testing the survey questionnaire and whether there were any modifications in the questionnaire after the pre-testing study or not. If any, what was the extent of the modifications? Please add this information in the methods section.
There is a lack of explanation concerning how the sample size has been calculated (i.e. what power and significance level?). This calculation of the sample size is very important when estimating and assessing the accuracy of the results.
Lines 157-159: I suggest deleting the statement “All participants 157 have been confirmed in accordance with the Declaration of Helsinki and have been approved by an appropriate ethics committee.” It is redundant.
Statistical analysis is too poorly described. It is not clear that two separate regression models were performed. Moreover, I suggest to better describe the criteria used for the model building strategy (i.e., choice of covariates to include in the model). For example, age was not included in the model. Is it correct?
Results
One of the major criticism is the comparison between two markedly different populations, as mentioned in the paragraph "limitations". The general population in the study is mostly middle-aged and equally distributed between men and women. On the other hand, in the HCW group the participants are mainly young women with university degrees and at high risk of contagion. Therefore, the higher approval of the viral vector in the HCW group (p = 0.056, Table 4) is difficult to interpret. To assess whether the viral vector is more acceptable in one group or the other, I suggest making a comparison through two regression models in which the "viral vector" category is compared with a reference category (e.g., no vaccine) and subsequently the two regression coefficients are compared. Alternatively, it should be emphasized in the "limitations" paragraph that the t-tests may lead to spurious conclusions.
Table 1: Percentages do not total 100. Please, carefully check.
Table 2: means and standard deviations are calculated as descriptive statistics. It is speculative that satisfaction items are approximatively normally distributed, since these categorical variables should be described by counts (n) and percentages (%).
Discussion
In order to generalize the study findings outside of Thailand, I suggest to add comparisons with some recent studies conducted in other countries.
I suggest to comment an important potential determinant of vaccine hesitancy, i.e. communication and media environment. Moreover, the role of health care providers as key components to improve the public trust to scientific and epidemiological evidence about vaccination. It is important to emphasized that health care providers could act as key components to overcome the public mistrust.
Please, comment and quote the following study: doi: 10.1016/j.ijnurstu.2022.104241 .
The Authors provided a comment about the role of academic knowledge in the satisfaction with different types of vaccines. I agree with this comment, and it would be important to stress the role of knowledge and education in implementing confidence in vaccines, quoting additional references regarding other studies conducted in different geographic areas (doi.org/10.1101/2021.06.16.21258808, doi.org/10.1371/journal.pone.0270684).
Considering the important epidemiological and economic burden associated with SARS-CoV-2 infection, I suggest making the results of the study more fruitful to propose priority actions for ameliorating vaccine acceptance and supporting decision-makers. Especially given the low uptake of the vaccine booster among the general population.
I suggest to add more details about actions implemented to overcome study limitations.
Lines 381-385: I suggest moving future research from Limitations section.
Author Response
Reviewer 1
Comments and Suggestions for Authors
The manuscript is intended to describe satisfaction with COVID-19 vaccines and boosters 129 among members of general population (GP) and healthcare workers (HCWs).In its current version, the study presents some flaws that must be addressed before it could be considered worthy of publication on Vaccines.
Following are revisions needed:
Introduction
The introduction should specify the theme, the context, the justification, and the hypotheses of the research.
Authors:
“The issue of general population (GP) and healthcare workers (HCWs) COVID-19 vac-cine booster might be affected by many things or harbored feelings such as positive/negative that refer to satisfaction, based on sociodemographic characteristics of the individuals. Thus, this study evaluated satisfaction with COVID-19 vaccines and boosters among members of GP and HCWs. We examined the effect of socio-demographic factors such as age, gender, status, education, occupation, risk of contracting COVID-19, and vac-cine and booster history and satisfaction scores in each group. To our knowledge, this is the only study to examine both of these groups in the urban community of Bangkok, Thailand.”
In the revised paper we have mention the following sentence in introduction.
This introductory part should end with the main objective of the study.
Authors:
The main objective we added in the last paragraph of introduction that “Thus, this study evaluated satisfaction with COVID-19 vaccines and boosters among members of GP and HCWs. We examined the effect of socio-demographic factors such as age, gender, status, education, occupation, risk of contracting COVID-19, and vaccine and booster history and satisfaction scores in each group”.
In the revised paper we have mention the following sentence in introduction.
The authors stated: “To-date, few studies have focused on satisfaction with the COVID-19 vaccine booster.”. Please, cite these studies.
Authors:
We would like to explain that:
The revised paper we have added the following sentences “To-date, few studies have focused on satisfaction with the COVID-19 vaccine booster [13-15]”.
The reference added:
- Gewirtz-Meydan, A., et al., COVID-19 Among Youth in Israel: Correlates of Decisions to Vaccinate and Reasons for Refusal. Journal of Adolescent Health, 2021.
- Symons, X., S. Matthews, and B. Tobin, Why should HCWs receive priority access to vaccines in a pandemic? BMC Medical Ethics, 2021. 22(1): p. 79.
- Nguyen, L.H., et al., Risk of COVID-19 among front-line health-care workers and the general community: a prospective cohort study. The Lancet Public Health, 2020. 5(9): p. e475-e483.
Methods
It is not given any information regarding a pilot study for testing the survey questionnaire and whether there were any modifications in the questionnaire after the pre-testing study or not. If any, what was the extent of the modifications? Please add this information in the methods section.
Authors:
The revised paper we have added the following sentences “Content validity was examined by three experts, and the item objective congruence (IOC) score was 0.93. This study reported a Cronbach’s alpha coefficient of 0.751” within the “questionnaire section”.
There is a lack of explanation concerning how the sample size has been calculated (i.e. what power and significance level?). This calculation of the sample size is very important when estimating and assessing the accuracy of the results.
Authors:
I have incorporated all of your suggestions into my revision. They were very helpful.
In 2021, National Statistical Office, The Government Complex Bangkok Thailand (http://www.nso.go.th/sites/2014/Pages/%E0%B8%9A%E0%B8%88/2562/04-62.aspx) reported that:
- GP approximately 7 million people.
- HCWs Approximately 4 hundred thousand people.
In addition, this study we focus on COVID-19 pandemic. This study was limited by a number of factors. For one, it was conducted online using self-reported data, which could introduce selection bias.
Thus, we would like to explain that to compute the number of participants by G*Power program
Determine: Effect size d = 0.236
∞ err prob = 0.05
Power (1-ᵦ err prob) = 0.95
Df = 778
Actual power = 0.95
Sample size group 1 = 390
Sample size group 2 = 390
Total sample size = 780
We hope you sympathize and agree with this.
Lines 157-159: I suggest deleting the statement “All participants 157 have been confirmed in accordance with the Declaration of Helsinki and have been approved by an appropriate ethics committee.” It is redundant.
Authors:
I have incorporated all of your suggestions into my revision. They were very helpful.
Thus, I removed sentences “All participants 157 have been confirmed in accordance with the Declaration of Helsinki and have been approved by an appropriate ethics committee” within section “Data collection”.
Statistical analysis is too poorly described. It is not clear that two separate regression models were performed. Moreover, I suggest to better describe the criteria used for the model building strategy (i.e., choice of covariates to include in the model). For example, age was not included in the model. Is it correct?
Authors:
I would like to explain that:
In multiple linear regression model. We are separated between GP and HCWs because; Satisfaction of individuals can be affected by many things or harbored feelings such as positive/negative, happiness and unhappiness in every stage of their lives, as a result of the situations they encounter, their satisfaction or dissatisfaction with life may change.
In model do not add age in the model because; GP age (maximum = 85 years old, minimum = 2 years old). HCWs age (maximum = 61 years old, minimum = 20 years old). “Very wide range”.
Results
One of the major criticism is the comparison between two markedly different populations, as mentioned in the paragraph "limitations".
Authors:
The revised paper we have added the following sentences “the comparison between two markedly different populations”. Was added within "limitations section”
The general population in the study is mostly middle-aged and equally distributed between men and women. On the other hand, in the HCW group the participants are mainly young women with university degrees and at high risk of contagion. Therefore, the higher approval of the viral vector in the HCW group (p = 0.056, Table 4) is difficult to interpret.
Authors:
I have incorporated all of your suggestions into my revision. They were very helpful.
Age was presented by continuous variable. I would like to explain that;
GP age (maximum = 85 years old, minimum = 2 years old).
HCWs age (maximum = 61 years old, minimum = 20 years old).
Therefore, is difficult to interpret but we type to explain that “our results indicate no significant difference in GP and HCWs participants’ satisfaction with any of the vaccine types; however, the difference in satisfaction with the viral vector vaccine (p = 0.056) was nearly significant”.
To assess whether the viral vector is more acceptable in one group or the other, I suggest making a comparison through two regression models in which the "viral vector" category is compared with a reference category (e.g., no vaccine) and subsequently the two regression coefficients are compared.
Authors:
I have incorporated all of your suggestions into my revision. They were very helpful.
I would like to presented that: We try to sum of “Vaccination history” including 1 – 3 doses of vaccination.
The reason to sum of “Vaccination history” because the first time of vaccination in Thailand has a lot of “Inactivated vaccine” for normal population, after that Thai government support to mRNA and Viral vector vaccine for normal population.
As the sample, if they do not want to wait for a vaccine from the government, they can subscribe for vaccine from private hospital.
At this moment, government have vaccine for all population in Thailand. Now a day can be should AZ or Pfizer. Big promote for young children and elderly, and promote for vaccine boosters also.
|
Vaccination history |
GP |
|
HCWs |
|
|
|
|
1st dose |
Ref. |
|
Ref. |
|
|
|
2nd dose |
-0.304 (-1.303, 0.694) |
0.549 |
-0.510 (-1.391, 0.371) |
0.256 |
|
|
3rd dose |
0.180 (-0.645, 1.005) |
0.668 |
-0.844 (-1.177, -0.512) |
< 0.001 |
The revised paper we have added the following in “table 5”.
Alternatively, it should be emphasized in the "limitations" paragraph that the t-tests may lead to spurious conclusions.
Authors:
The revised paper we have added the following sentences “In addition, this study we used t-test analysis might be lead to spurious conclusions” added within "limitations section”
Table 1: Percentages do not total 100. Please, carefully check.
Authors:
I have incorporated all of your suggestions into my revision. They were very helpful.
The revised paper we have added the following in “table 1”.
Table 2: means and standard deviations are calculated as descriptive statistics. It is speculative that satisfaction items are approximatively normally distributed, since these categorical variables should be described by counts (n) and percentages (%).
Authors:
The satisfaction section consists of 15 items scored as very satisfied = 3, medium satisfied = 2, and low satisfied = 1. Satisfaction item responses were summed to a total score ranging from 0–45 and classified into three categories, namely low satisfaction (15–25), medium satisfaction (26–35), and high satisfaction (36–45) (see Table 2 for details).
Explains that;
“Low satisfaction” (score ranging 15–25) represent to scaled “1”.
“Medium satisfaction” (score ranging 26–35) represent to scaled “2”.
“High satisfaction” (score ranging 36–45) represent to scaled “3”.
Thus, satisfaction scaled 0, 1, 3 were classified into three categories for represent satisfaction item responses
In addition, in “Table 2” we presented by “sum score of satisfaction”.
Thus, in table 2 were showed means and standard deviations are calculated as descriptive statistics.
Discussion
In order to generalize the study findings outside of Thailand, I suggest to add comparisons with some recent studies conducted in other countries.
Authors:
I have incorporated all of your suggestions into my revision. They were very helpful.
In addition, we added reference;
“Which among the HCPs, with higher scores being associated with mandatory vaccination support (p < 0.001) [31].”
Additionally, social determinants of health such as gender, age, number of children in the family, and the degree of satisfaction with life were also predictors (p < 0.05) for COVID-19 vaccine acceptance [32].
One study presented that satisfaction with vaccine-related promotional materials were determinants of behavioral intention [33].
Other study presented that 70% intended to be vaccinated when the COVID-19 vac-cine was approved under Emergency Use Administration [34].
- Giannakou K, Kyprianidou M, Christofi M, Kalatzis A, Fakonti G. Mandatory COVID-19 Vaccination for Healthcare Professionals and Its Association With General Vaccination Knowledge: A Nationwide Cross-Sectional Survey in Cyprus. Frontiers in Public Health. 2022;10.
- Fobiwe JP, Martus P, Poole BD, Jensen JL, Joos S. Influences on Attitudes Regarding COVID-19 Vaccination in Germany. Vaccines. 2022;10(5).
- Zhang K, Fang Y, Chan PSF, Cao H, Chen H, Hu T, et al. Behavioral Intention to Get a Booster Dose of COVID-19 Vaccine among Chinese Factory Workers. International Journal of Environmental Research and Public Health. 2022;19(9).
- Yang J, Liao Y, Hua Q, Lv H. A Survey of Awareness of COVID-19 Knowledge, Willingness and Influencing Fac-tors of COVID-19 Vaccination. Vaccines. 2022;10(4).
- Giannakou K, Kyprianidou M, Heraclides A. Attitudes and Determinants of Mandatory Vaccination against COVID-19 among the General Population of Cyprus: A Nationwide Cross-Sectional Study. Vaccines. 2022;10(3).
I suggest to comment an important potential determinant of vaccine hesitancy, i.e. communication and media environment. Moreover, the role of health care providers as key components to improve the public trust to scientific and epidemiological evidence about vaccination. It is important to emphasized that health care providers could act as key components to overcome the public mistrust.
Please, comment and quote the following study: doi: 10.1016/j.ijnurstu.2022.104241 .
Authors:
The important that healthcare professionals and representative groups, should be closely involved in policymaker and health authority decisions regarding the establishment and implementation of vaccine recommendations and interventions to address vaccine hesitancy [36].
In addition, we added reference
- Peters MDJ. Addressing vaccine hesitancy and resistance for COVID-19 vaccines. Int J Nurs Stud. 2022;131:104241.
The Authors provided a comment about the role of academic knowledge in the satisfaction with different types of vaccines. I agree with this comment, and it would be important to stress the role of knowledge and education in implementing confidence in vaccines, quoting additional references regarding other studies conducted in different geographic areas (doi.org/10.1101/2021.06.16.21258808, doi.org/10.1371/journal.pone.0270684).
Authors:
However, for people with high level of stress, other health programs need to be developed to enhance their positive attitude toward the COVID-19 vaccine [37].
They rated fear of side effects (87.18%) and lack of information (70.94%) as the most rea-sons for the hesitancy [38].
In addition, we added reference;
- Zhang H, Li Y, Peng S, Jiang Y, Jin H, Zhang F. The effect of Health Literacy on COVID-19 Vaccine Hesitancy: The Moderating Role of Stress. medRxiv. 2021:2021.06.16.21258808.
- Rahman MM, Chisty MA, Alam MA, Sakib MS, Quader MA, Shobuj IA, et al. Knowledge, attitude, and hesitancy towards COVID-19 vaccine among university students of Bangladesh. PLoS One. 2022;17(6):e0270684.
Considering the important epidemiological and economic burden associated with SARS-CoV-2 infection, I suggest making the results of the study more fruitful to propose priority actions for ameliorating vaccine acceptance and supporting decision-makers. Especially given the low uptake of the vaccine booster among the general population.
Authors:
Thank you for suggestion
I suggest to add more details about actions implemented to overcome study limitations.
Authors:
This study was limited by a number of factors. For one, it was conducted online using self-reported data, which could introduce selection bias. Therefore, this sampling method can have a potential sampling bias, margin of error, and the comparison between two markedly different populations. In addition, this study we used t-test analysis might be lead to spurious conclusions. Finally, our study was unbalanced in terms of sex distribution that could introduce conclusion bias.
Lines 381-385: I suggest moving future research from Limitations section.
Authors:
The revised paper we have moving future research from Limitations section.

Reviewer 2 Report
In this publication the authors describe the satisfaction levels of a sample of individuals regarding COVID-19 vaccination campaign. The text is easy to ready and follow, on an interesting and relevant topic, which is evaluation of vaccination campaigns. However, there are certain aspects that need to be reviewed by the authors:
- I would suggest to reduce the extension of the "Introduction" with a more condensed and straight-to-the-point structure: 1) What do you know?, 2) What do you not know?, 3) What have you done?
- Please, be consistent with abbreviation and acronym use: "coronavirus disease 2019 (COVID-19)", not "COVID-19 (SARS-CoV-2)", for example. Introduce each abbreviation or acronym the very first time they are used.
- Could you make a figure with the promotion tools and activities for your data collection? That could be of great help for similar studies in the future.
- Based on your answer options, I would say that you asked for the sex rather than gender. Please, stick to the correct term as needed (sex=biology, gender=sociology).
- Please, use median and interquartile range for continuous variables in case these are non-parametric or non-normally distributed (which I guess it is the case). Additionally, it would be interesting to know also the minimum and the maximum for these variables.
- Also related to variable distribution, if this was non-normal or non-parametric, please use the appropriate comparison tests: Mann-Whitney U and Kruskal-Wallis H.
- How long was the survey open for answers?
- Why are there so big differences between GP and HCW in: age, sex?
- How is evaluated the lack of risk for COVID-19 acquisition?
- Could you describe how was the COVID-19 vaccination campaign in Thailand? Did the patients have freedom to select the type of vaccine? Did they have to pay for it? Was there any guidance regarding the type of patient (age for example) and the type of vaccine to be administered (AZ and J&J only for elderly, heterologous vaccination...)?
- If there is a direct correlation between the score obtain with your survey and the satisfaction of the patient... What happens when a patient does not get a viral vector vaccine (question 2)? They have 0 points for that question? Then that hampers the overall score for this patient. Could you provide details and explain whether this situation is possible? Can all your individuals have a reply to all the questions?
- Why do you provide Table 3 separate from Table 2? Are not the values in Table 3 the results of the summary of all individual values in Table 2?
- Table 4: Why mRNA vaccines are only an option for 3rd dose?
- Are GP and HCW populations comparable? How did you check this?
- You state "To the best of our knowledge, it is the first study to evaluate satisfaction with the COVID-19 vaccine booster among both GP participants and HCWs in an urban community. " However, later, in the following paragraph you mention that there are previous experiences in HCW. Considering that the data for GP and HCW are presented always separately, I am not sure whether your statement is 100% accurate.
- "vaccine allocation was distributed across all areas" If your sample is from urban areas, how can they evaluate adequately the vaccine distribution in non-urban areas? Is that judgement reliable?
- Considering the results of Table 2, I have the feeling that your result interpretation is a bit too optimistic, specially regarding questions #1, #11, and #15, where the score is below 1.50 (50% of 3, the maximum score).
Author Response
Reviewer 2
Comments and Suggestions for Authors
In this publication the authors describe the satisfaction levels of a sample of individuals regarding COVID-19 vaccination campaign. The text is easy to ready and follow, on an interesting and relevant topic, which is evaluation of vaccination campaigns. However, there are certain aspects that need to be reviewed by the authors:
Authors:
I have incorporated all of your suggestions into my revision. They were very helpful.
- I would suggest to reduce the extension of the "Introduction" with a more condensed and straight-to-the-point structure: 1) What do you know?, 2) What do you not know?, 3) What have you done?
What do you know?
Authors:
Pandemic outbreaks and vaccine inaccessibility are major challenges to achieving 100% vaccination rates. In addition, distrust of vaccine boosters has been triggered by anti-vaccination campaigns and is becoming increasingly prevalent. Epidemiologists and public health researchers are experiencing challenges in their efforts to provide evidence informed recommendations regarding COVID-19 vaccine booster timing and pro-mote full vaccination to priority groups. Individuals who have already been vaccinated may accept or reject a vaccine booster dose due to various reasons, including the side-effects they experienced following previous (primer) doses, perceived ineffectiveness of the vaccine booster dose, susceptibility to the target infection, and safety uncertainties.
What do you not know?
Authors:
The factors influenced on satisfaction included the sociodemographic characteristics of the individuals, individual factors, social and organizational factors. In addition, certain characteristics of COVID-19 vaccines themselves influenced public attitudes towards accepting the vaccines. Attitudes towards vaccines are influenced by a complex interaction of various social, cultural, political, and personal factors, including doubts and hesitation about vaccination. It is widely believed that factors such as risk perception beliefs have a substantial effect on the intention to be vaccinated. However, individuals who have already been vaccinated may accept or reject a vaccine booster dose due to various reasons, including side effects experienced following the previous (primer) doses, the perceived effectiveness of the vaccine booster dose, the vaccinee’s susceptibility to the tar-get infection, and other safety concerns.
Thus, few studies have focused on satisfaction with the COVID-19 vaccine booster. The issue of general population (GP) and healthcare workers (HCWs) COVID-19 of the vaccine booster might be affected by many things or harbored feelings such as positive or negative that refer to satisfaction and based on sociodemographic characteristics of the individuals. Thus, this study evaluated satisfaction with COVID-19 vaccines and boosters among members of GP and HCWs. We examined the effect of socio-demographic factors such as age, gender, status, education, occupation, risk of contracting COVID-19, and vac-cine and booster history and satisfaction scores in each group. To our knowledge, this is the only study to examine both of these groups in the urban community of Bangkok, Thailand
What have you done?
Authors:
- HCWs had significantly higher total satisfaction scores than GP participants (p = 0.034), and they were also significantly more satisfied with the mRNA vaccine as a third dose (p = 0.001).
- Multiple linear regression models found less association with vaccine satisfaction among GP participants who had not isolated following exposure to COVID-19 and those who have never been at risk of infection (ᵦ -0.159; 95% CI -12.867, -1.877; p = 0.009).
- Among HCWs, being married (ᵦ 0.157; 95% CI 0.794, 3.278; p = 0.001) or divorced (ᵦ 0.198; 95% CI 3.303, 9.596; p <0.01) was more closely associated with vaccine satisfaction than being single.
HCWs were more satisfied with the type and efficacy of inactivated, viral vector, and mRNA vaccines than GP participants, and the former were also more satisfied with the cost of vaccine boosters.
- Our results indicate that satisfaction with the COVID-19 vaccine is based on academic knowledge sharing and the government’s promotion efforts.
- Future research will explore strategies to raise awareness about the importance of vaccination.
- Please, be consistent with abbreviation and acronym use: "coronavirus disease 2019 (COVID-19)", not "COVID-19 (SARS-CoV-2)", for example. Introduce each abbreviation or acronym the very first time they are used.
Authors:
The revised paper we have added the following sentences in “Abbreviations” and in all text.
Coronavirus disease 2019 (COVID-19); BMI: Body Mass Index; GP: (general population); HCWs: (healthcare workers); severe acute respiratory syndrome coronavirus-2 (SARS-CoV-2); messenger ribonucleic acid (mRNA); WHO: World Health Organization; β: Beta (Standardized coefficients; CI: Confidence Interval; r: correlation coefficient
- Could you make a figure with the promotion tools and activities for your data collection? That could be of great help for similar studies in the future.
Authors:
I have incorporated all of your suggestions into my revision. They were very helpful.
The revised paper we have added the following “Figure 1” in study design.
- Based on your answer options, I would say that you asked for the sex rather than gender. Please, stick to the correct term as needed (sex=biology, gender=sociology).
Authors:
I have incorporated all of your suggestions into my revision. They were very helpful.
Thus. I change from “gender’ to “sex”
- Please, use median and interquartile range for continuous variables in case these are non-parametric or non-normally distributed (which I guess it is the case). Additionally, it would be interesting to know also the minimum and the maximum for these variables.
Authors:
I have incorporated all of your suggestions into my revision. They were very helpful.
Thus, I change from “mean” to “median” added into table 1.
Age was presented by continuous variable. I would like to explain that;
GP age (maximum = 85 years old, minimum = 2 years old).
HCWs age (maximum = 61 years old, minimum = 20 years old).
- How long was the survey open for answers?
Authors:
4 months from September to December 2021.
- Why are there so big differences between GP and HCW in: age, sex?
Authors:
I would like to explain that
- HCWs were presented by who age more that 18 years old. In addition, were showed female than male because most of all are nurse.
- GP were presented by normal population, which were showed vary age and sex.
- How is evaluated the lack of risk for COVID-19 acquisition?
Authors:
We ask about do you are risk of contracting COVID-19. Devised into 4 categories including;
- Risk exposure: isolation
- Risk exposure: no isolation
- Not sure
- Never at risk
In addition, definition of “Risk” mean that who contact or closely with someone COVID-19 infection.
- Could you describe how was the COVID-19 vaccination campaign in Thailand? Did the patients have freedom to select the type of vaccine? Did they have to pay for it? Was there any guidance regarding the type of patient (age for example) and the type of vaccine to be administered (AZ and J&J only for elderly, heterologous vaccination...)?
Authors:
I would like to explain base on real situation as the first time for promote vaccination in Thailand. As the 1st dose vaccination to HCWs because GP are concerned about the safety of vaccine. Thus, the Thai government has campaigned for the HCWs to vaccinated first, for confirm safety of vaccine.
As the first time of vaccination in Thailand has a lot of “Inactivated vaccine” for normal population, after that Thai government support to mRNA and Viral vector vaccine for normal population.
As the sample, if they do not want to wait for a vaccine from the government, they can subscribe for vaccine from private hospital.
At this moment, government have vaccine for all population in Thailand. Now a day can be should AZ or Pfizer. Big promote for young children and elderly, and promote for vaccine boosters also.
- If there is a direct correlation between the score obtain with your survey and the satisfaction of the patient... What happens when a patient does not get a viral vector vaccine (question 2)? They have 0 points for that question? Then that hampers the overall score for this patient. Could you provide details and explain whether this situation is possible? Can all your individuals have a reply to all the questions?
Authors:
In would like to explain that:
For the satisfaction score we presented by “sum score” or “total score”
Thus, if who does not get vaccination or showed 0 points, we are reply to all the questions also.
- Why do you provide Table 3 separate from Table 2? Are not the values in Table 3 the results of the summary of all individual values in Table 2?
Authors:
I would like to explain that:
In table 2, would like presented with descriptive statistics of satisfaction scores on COVID-19 vaccine among GP and HCWs, in “each satisfaction” items including vaccination booster type, vaccination booster management, satisfactions with vaccine booster.
In addition, in table 3 we would like to presented about comparison of total vaccine satisfaction scores, by “total satisfaction score” (that separated from table 2). We think that if separated might be clear and more understand.
Moreover, Table 3 presents the result of the independent t-test comparing total vaccine satisfaction scores between GP and HCWs participants. The table shows that HCWs expressed significantly higher satisfaction with the COVID-19 vaccine booster than GP participants (p = 0.037).
- Table 4: Why mRNA vaccines are only an option for 3rd dose?
Authors:
I would to explain that;
At the 1st dose vaccination in Thailand don’t have mRNA vaccine for population. After that, there can be choice of the vaccine type.
In the beginning, there were still limitations of the lack of vaccines. Now day, population in Thailand can be choice of the vaccine type.
- Are GP and HCW populations comparable? How did you check this?
Authors:
I would like to explain that:
Satisfaction is the pleasure that is felt when you do something or get something done or it can be defined as emotional responses to life consisting of work, leisure time and other non-work time.
Satisfaction of individuals can be affected by many things or harbored feelings such as positive/negative, happiness and unhappiness in every stage of their lives, as a result of the situations they encounter, their satisfaction or dissatisfaction with life may change.
Thus, we think that GP and HCW populations can comparable because individuals who have already been vaccinated may accept or reject a vaccine booster dose due to various reasons, including side effects experienced following the previous (primer) doses, the perceived effectiveness of the vaccine booster dose, the vaccinee’s susceptibility to the target infection, and other safety concerns.
In addition, we would like to know that between GP and HCWS they think the same or difference on satisfaction with COVID-19 vaccines.
Form our results found that HCWs were more satisfied with vector vaccine than GP participants and had good knowledge and understanding of vaccines that have been determined to be of good quality. From these results if promote knowledge to GP might be understanding of vaccines like HCWs.
- You state "To the best of our knowledge, it is the first study to evaluate satisfaction with the COVID-19 vaccine booster among both GP participants and HCWs in an urban community. " However, later, in the following paragraph you mention that there are previous experiences in HCW. Considering that the data for
Authors:
I would like to explain that:
“In addition, we found that HCWs were significantly more satisfied with vaccines overall as well as with mRNA vaccines as a third dose”. Mean that HCWs within this study has experiences for vaccine more than GP.
- "vaccine allocation was distributed across all areas" If your sample is from urban areas, how can they evaluate adequately the vaccine distribution in non-urban areas? Is that judgement reliable?
Authors:
I would like to explain that:
The population in urban community have a lot of population types such as education and income. At the same time pandemic distribution in urban than non-urban areas. From this evident lend to the reason why we focus in urban community.
- Considering the results of Table 2, I have the feeling that your result interpretation is a bit too optimistic, specially regarding questions #1, #11, and #15, where the score is below 1.50 (50% of 3, the maximum score).
Authors:
I would like to explain that:
The second section consisted of 15 items measuring satisfaction, which were scored on a 3-point Likert scale ranging from
- High satisfaction = 3,
- Medium satisfaction = 2,
- Low satisfaction = 1
In Q1. (I am satisfied with inactivated vaccine = 1.25±0.92 and 1.39±0.60), mean that GP and HCWs might be “low satisfied” with inactivated vaccine.
In Q11. (The government’s credibility in procuring and administering vaccines = 1.48±1.16 and 1.43±0.60), mean that GP and HCWs might be “low satisfied” with government’s credibility in procuring and administering vaccines.
In Q15. (15. I am satisfied with the vaccine cost = 1.45±0.95), mean GP might be “low satisfied” but HCWs “medium satisfied”

Reviewer 3 Report
The manuscript, "Satisfaction with COVID-19 vaccines in healthcare workers and the general population: a cross sectional study in urban Bangkok, Thailand" is touching on an important topic of global importance. With the connected world, the vaccination coverage is critical for controlling the speed and scale of the COVID-19 pandemic. Important and relevant for all and every country and people.
The findings presented in the study is very important, relevant and has important take home messages. While, in general, I appreciate the findings, few suggestions may be taken into account for enhanced clarity. This includes,
1) The difference of vaccine type usage between the GP and HCW is significantly highlighted. I am not sure whether the choice of the vaccine type, between mRNA vaccine and the inactivated vaccine, was available for both the GP and the HCW.
2) Was it a Informed choice for the groups - GP and HCW? OR it is based on availability for the HCW and the GP?
3) Thus, it may be important to see how the vaccine was rolled out for usage in Thailand. World over, the vaccines were rolled in countries based on availability.
4) Especially important is prioritization of the HCW for vaccination. Is it that the differential usage is based on availability and prioritization of vaccination between the HCW followed by GP.
I am ok with rest of the findings.
Author Response
Reviewer 3
Comments and Suggestions for Authors
The manuscript, "Satisfaction with COVID-19 vaccines in healthcare workers and the general population: a cross sectional study in urban Bangkok, Thailand" is touching on an important topic of global importance. With the connected world, the vaccination coverage is critical for controlling the speed and scale of the COVID-19 pandemic. Important and relevant for all and every country and people.
The findings presented in the study is very important, relevant and has important take home messages. While, in general, I appreciate the findings, few suggestions may be taken into account for enhanced clarity. This includes,
Authors:
I have incorporated all of your suggestions into my revision. They were very helpful.
1) The difference of vaccine type usage between the GP and HCW is significantly highlighted. I am not sure whether the choice of the vaccine type, between mRNA vaccine and the inactivated vaccine, was available for both the GP and the HCW.
Authors:
I would like to explain base on real situation as the first time for promote vaccination in Thailand. As the 1st dose vaccination to HCW because GP are concerned about the safety of vaccine. Thus, the Thai government has campaigned for the HCW to vaccinate first, for confirm safety of vaccine.
As the first time of vaccination in Thailand has a lot of inactivated vaccine for normal population, later after that Thai government support to mRNA and Viral vector vaccine for normal population.
Within this study the research population cannot choice of the vaccine type. Now day, population in Thailand can be choice of the vaccine type.
2) Was it a Informed choice for the groups - GP and HCW ? OR it is based on availability for the HCW and the GP?
Authors:
During on did research population in Thailand can not choice of the vaccine type but now a day based on availability for the HCW and the GP.
3) Thus, it may be important to see how the vaccine was rolled out for usage in Thailand. World over, the vaccines were rolled in countries based on availability.
Authors:
In the beginning, there were still limitations of the lack of vaccines. Now day, population in Thailand can be choice of the vaccine type.
4) Especially important is prioritization of the HCW for vaccination. Is it that the differential usage is based on availability and prioritization of vaccination between the HCW followed by GP.
Authors:
Thank you for your comment and suggestion. Many people are concerned about the potential side-effects of vaccines, and COVID-19 vaccination hesitancy has become a global concern including GP and HCW.
Like previously explain that the 1st dose vaccination to HCW because GP are concerned about the safety of vaccine. Thus, the Thai government has campaigned for the HCW to vaccinate first, for confirm safety of vaccine. Now day, population in Thailand can be choice of the vaccine type

This manuscript is a resubmission of an earlier submission. The following is a list of the peer review reports and author responses from that submission.
Round 1
Reviewer 1 Report
I congratulate the Authors of the manuscript for their work.
This interesting study investigates satisfaction with COVID-19 vaccines among healthcare professionals and the general population in a convenience sample.
I find the satisfaction analysis interesting through the analysis of the type of vaccine received and of different socio-demographic characteristics.
Before publication, there are some clarifications that mainly concern the methodology that could, once clarified, make the evidence generated even more solid, in my opinion.
A first consideration concerns the convenience sample.
If I understand correctly, the authors have provided for the calculation of an adequate sample size a priori through software. It would be interesting to state what parameters were used to calculate this data: the error value, the number of residents in Bangkok, the number of healthcare workers in Bangkok? Please the authors, if possible, to clarify this element.
Still with regard to materials and methods, I would have another consideration to make. The use of indicators such as the mean and standard deviation requires that the distribution of linear variables have a normal distribution. It is necessary to specify that it has been ascertained whether the age or, for example, the values ​​of each item indicated have a normal distribution with specific tests (kurtosis?).
When it comes to using linear regression, there are also some considerations to make. In fact, some assumptions must be demonstrated and verified a priori with specific tests. Multiple regression assumes that the residuals are normally distributed and that the independent variables are not highly correlated with each other (VIF?).
I kindly ask the authors to clarify these methodological aspects which I consider important for the validity of the results reported.
Author Response
Reviewer 1
This interesting study investigates satisfaction with COVID-19 vaccines among healthcare professionals and the general population in a convenience sample. I find the satisfaction analysis interesting through the analysis of the type of vaccine received and of different socio-demographic characteristics. Before publication, there are some clarifications that mainly concern the methodology that could, once clarified, make the evidence generated even more solid, in my opinion.
Authors
I have incorporated all of your suggestions into my revision. They were very helpful.
……………….
A first consideration concerns the convenience sample. If I understand correctly, the authors have provided for the calculation of an adequate sample size a priori through software. It would be interesting to state what parameters were used to calculate this data: the error value, the number of residents in Bangkok, the number of healthcare workers in Bangkok? Please the authors, if possible, to clarify this element.
Authors
I have incorporated all of your suggestions into my revision. They were very helpful.
In 2021, National Statistical Office, The Government Complex Bangkok Thailand (http://www.nso.go.th/sites/2014/Pages/%E0%B8%9A%E0%B8%88/2562/04-62.aspx) reported that:
- GP approximately 7 million people.
- HCWs Approximately 4 hundred thousand people.
In addition, this study we focus on COVID-19 pandemic. This study was limited by a number of factors. For one, it was conducted online using self-reported data, which could introduce selection bias.
Thus, we would like to explain that to compute the number of participants by G*Power program
Determine: Effect size d = 0.236
∞ err prob = 0.05
Power (1-ᵦ err prob) = 0.95
Df = 778
Actual power = 0.95
Sample size group 1 = 390
Sample size group 2 = 390
Total sample size = 780
We hope you sympathize and agree with this.
……………….
Still with regard to materials and methods, I would have another consideration to make. The use of indicators such as the mean and standard deviation requires that the distribution of linear variables have a normal distribution. It is necessary to specify that it has been ascertained whether the age or, for example, the values ​​of each item indicated have a normal distribution with specific tests (kurtosis?).
Authors
We test normal distribution, presented by Kolmogorov-Smirnov test were showed p-value > 0.05.
Thus, this study used linear regression model for analysis.
……………….
When it comes to using linear regression, there are also some considerations to make. In fact, some assumptions must be demonstrated and verified a priori with specific tests. Multiple regression assumes that the residuals are normally distributed and that the independent variables are not highly correlated with each other (VIF?).
Authors
Multiple linear regression was used to identify predictors of satisfaction scores among GP and HCWs.
Independent variable including gender, status, education, and risk of contracting COVID-19.
In addition, we test normal distribution before linear regression model analysis.
“Our results found that Among GP participants, who had been exposed to the risk of contracting COVID-19 but had not self-isolated was less associated with vaccine satisfaction than “never at risk” (ᵦ -0.159; 95% CI -12.867, -1.877; p = .009).”
“Among HCWs, being married (ᵦ 0.157; 95% CI 0.794, 3.278; p = .001) and or divorced (ᵦ 0.198; 95% CI 3.303, 9.596; p < 0.01) was more closely associated with vaccine satisfaction than being single. In addition, risk infection by not isolated of contracting COVID-19 (ᵦ -0.206; 95% CI -5.013, -0.242; p = .031) and those who had isolated after exposure to the COVID-19 (ᵦ -0.282; 95% CI -5.627, -0.990; p = .005) were less associated with vaccine satisfaction than “never at risk.””
In the revised paper we have mention the following sentence in result “Association between participant characteristics and satisfaction with COVID-19 vaccines” as Table 5.
……………….
I kindly ask the authors to clarify these methodological aspects which I consider important for the validity of the results reported.
Authors
“Content validity was examined by three experts, and the item objective congruence (IOC) score was 0.93. This study reported a Cronbach’s alpha coefficient of 0.751.”
“The questions were designed and modified by an expert team of researchers.”
In the revised paper we have mention the following sentence in “2.4. Questionnaire”.
We hope you sympathize and agree with this.
……………….

Reviewer 2 Report
Reviewed paper provides a discussion of satisfaction with the 15 COVID-19 vaccine in the general population (GP) and healthcare workers (HCWs) in Bangkok. Overall, the article is well structured and coherent and generally written in an appropriate manner. The abstract reflects the content a is stated. The research study methods are appropriate. The article is written clearly, but the article needs improvements • a lack of hypothesis, I don’t know what Authors wanted to proves • its need to explain (by literature) how Authors understand satisfaction ? which factors influenced on satisfaction ? • its need to add a theoretical part eg. literature review • its need in conclusion define the theoretical and practical implications of the research resultsAuthor Response
Reviewer 2
Reviewed paper provides a discussion of satisfaction with the 15 COVID-19 vaccine in the general population (GP) and healthcare workers (HCWs) in Bangkok.
Overall, the article is well structured and coherent and generally written in an appropriate manner.
The abstract reflects the content a is stated. The research study methods are appropriate.
The article is written clearly, but the article needs improvements
- a lack of hypothesis, I don’t know what Authors wanted to proves
Authors
I have incorporated all of your suggestions into my revision. They were very helpful.
“The issue of general population (GP) and healthcare workers (HCWs) COVID-19 vac-cine booster might be affected by many things or harbored feelings such as positive/negative that refer to satisfaction, based on sociodemographic characteristics of the individuals. Thus, this study evaluated satisfaction with COVID-19 vaccines and boosters among members of GP and HCWs. We examined the effect of socio-demographic factors such as age, gender, status, education, occupation, risk of contracting COVID-19, and vac-cine and booster history and satisfaction scores in each group. To our knowledge, this is the only study to examine both of these groups in the urban community of Bangkok, Thailand.”
In the revised paper we have mention the following sentence in introduction.
……………….
- its need to explain (by literature) how Authors understand satisfaction?
Authors
“Satisfaction is the pleasure that feel when do something or get something or can be defined as emotional responses to life consisting of work, leisure time and other non-work time [2]. Satisfaction of individuals can be affected by many things or harbored feelings such as positive/negative, happiness and unhappiness in every stage of their lives, ss a result of the situations they encounter, their satisfaction or dissatisfaction with life may change [29].”
In the revised paper we have mention and added the following sentence in introduction.
In addition, we added reference also “2) Kilic, M., N. Ustundag Ocal, and G. Uslukilic, The relationship of Covid-19 vaccine attitude with life satisfaction, religious attitude and Covid-19 avoidance in Turkey. Hum Vaccin Immunother, 2021. 17(10): p. 3384-3393.” and “29) Lavallee LF., et al., Development of the contentment with life assessment scale (CLAS): Using daily life experiences to verify levels of self-reported life satisfaction. Social Indicators Research. 2007;83(2):201-44.”
……………….
which factors influenced on satisfaction?
Authors
“Thus, factors influenced on satisfaction included the sociodemographic characteristics of the individuals, individual factors, social and organizational factors. In addition, certain characteristics of COVID-19 vaccines themselves influenced public attitudes towards accepting the vaccines [30].”
In the revised paper we have added the following sentence in introduction.
In addition, we added reference also “30) Al-Jayyousi GF., et al., Factors Influencing Public Attitudes towards COVID-19 Vaccination: A Scoping Review Informed by the Socio-Ecological Model. Vaccines (Basel). 2021;9(6):548.”
……………….
- its need to add a theoretical part eg. literature review
Authors
“In addition, cultural theories about life such as the Taoist view argued that good and bad things happen in life, have a significant influence over people’s global ratings of their life but not over their day-to-day experiences of life [29]. “
In the revised paper we have added the following sentence in introduction.
In addition, we added reference also “29) Lavallee LF., et al., Development of the contentment with life assessment scale (CLAS): Using daily life experiences to verify levels of self-reported life satisfaction. Social Indicators Research. 2007;83(2):201-44.”
……………….
- its need in conclusion defines the theoretical and practical implications of the research results
Authors
I have incorporated all of your suggestions into my revision. They were very helpful.
“However, satisfaction with COVID-19 vaccines and boosters among members of GP and HCWs might be based on individual factors such as gender, status, education, and risk of contracting COVID-19. Thus, this study is based on the current situation of GP and HCWs during the COVID-19 vaccines and boosters in Bangkok Thailand.”
In the revised paper we have added the following sentence in Conclusions.
We hope you sympathize and agree with this.
……………….

Round 2
Reviewer 2 Report
The article might be published.